# Using supermarket loyalty card data to investigate seasonal variation in laxative purchases in the UK

Romana Burgess[1,2]\*, Neo Poon[1,2,3], Edward Sloan[1], James Goulding[3], Helen Bould[2,4,5], Anya Skatova[1,2]

1 Digital Footprints Lab, Population Health Sciences, Bristol Medical School, University of Bristol, United Kingdom, 2 The MRC Integrative Epidemiology Unit at the University of Bristol, Population Health Sciences, Bristol Medical School, University of Bristol, United Kingdom, 3 N/Lab, University of Nottingham, Nottingham, United Kingdom, 4 Centre for Academic Mental Health, University of Bristol, Bristol, United Kingdom, 5 Gloucestershire Health and Care NHS Foundation Trust, Gloucestershire, United Kingdom

\* romana.burgess@bristol.ac.uk

## Abstract

While laxatives are designed to manage the symptoms of constipation, they are also known to be misused for weight management, particularly by individuals with eating disorders. This study investigates the relationship between laxative purchases and weight management by examining seasonal trends. Using real-world loyalty card data from a major UK pharmacy retailer spanning December 2013 to December 2014, we analyse self-medication purchasing patterns from 748,375 buyers to explore potential links with weight management behaviours. In pre-registered analysis, we use regression models to investigate our hypotheses: (1) the number of doses purchased would be greater in January compared to the December prior, reflecting motivations in relation to "New Year's resolutions" around weight loss, and (2) doses purchased would be greater in May-August compared to the subsequent September, reflecting an increased focus on body image during the summer. We examine differences between stimulant and non-stimulant laxatives, as stimulants are more commonly misused for weight control due to their rapid effects. To validate our findings, we compare purchasing patterns with those for weight management products over the same periods, and also include negative controls of unrelated products, including painkillers, cold and flu, hay fever, and shampoo. Our findings reveal seasonal variations in laxative purchasing, particularly for non-stimulant medications. Purchases increase in January compared to December and are higher in some summer months compared to September, which may be consistent with seasonal patterns in weight-related behaviours. Non-stimulants exhibit greater seasonal fluctuation than stimulants. Purchases of weight management products follow similar patterns, aligning with established seasonal trends in weight loss behaviours. While laxative purchase trends align with those of weight management products, these patterns provide only indirect evidence and cannot confirm underlying intent, like body image concerns. This work

**Data availability statement:** The data underlying this study consist of pseudonymised supermarket loyalty card transactions provided by a leading UK pharmacy retailer under a data-sharing partnership with the University of Nottingham. Due to legal and commercial confidentiality restrictions, the raw dataset cannot be shared publicly. Researchers interested in accessing the data under equivalent conditions should contact the N/Lab Team at the University of Nottingham via nlab@notting-ham.ac.uk, who can provide guidance on the process for applying to the retailer for access. All analysis code, product categorisation lists, dosage conversion scripts, and realistic synthetic data can be accessed via our GitHub repository (https://github.com/RomanaBurgess/loyalty-card-laxatives/).

**Funding:** A.S is supported by a UKRI Future Leaders Fellowship (MR/T043520/1; https://www.ukri.org/) and an ESRC Smart Data Accelerator Award (ES/Y010973/1; https://www.ukri.org/councils/esrc/). H.B. is funded by an NIHR Advanced Fellowship (302271; https://www.nihr.ac.uk/). The funders had no role in study design, data collection and analysis, decision to publish, or preparation of the manuscript.

**Competing interests:** The authors have declared that no competing interests exist.

highlights an opportunity for loyalty card data to evaluate impacts of policy regulations in a real-world setting.

---

## Author summary

Laxatives are commonly used to relieve constipation, but they can also be misused by people trying to lose weight, particularly those struggling with eating disorders. We wanted to understand whether people in the UK might be using laxatives for weight control at certain times of the year. To do this, we analysed loyalty card data from a large UK pharmacy to look at when people bought laxatives over a one-year period. We found that purchases of laxatives increased during the summer months, a time when many people focus on their appearance and may try to lose weight because of the warmer weather. We also saw similar patterns in purchases of diet-related products, which supports the idea that laxatives may sometimes be used for weight management. Our research shows how loyalty card data can offer insights into health behaviours in the general population. These findings could help inform public health efforts and policy, especially around eating disorders and the regulation of over-the-counter medications. This approach could also be applied to other types of health-related behaviours, offering a new way to support early identification and intervention.

## 1. Introduction

Laxatives are medications used to manage constipation, readily available to buy over-the-counter in the UK. The prevalence of the use of such medication varies widely, ranging from 1% to 18% in the general population and from 3% to 59% among those experiencing constipation [1]. However, laxatives are also commonly misused by individuals with eating disorders as an unhealthy weight-control behaviour [2], with clinical studies reporting usage rates as high as one in four [3,4]. This widespread misuse poses significant public health concerns [5–7]. Particularly alarming is the misuse of stimulant laxatives, which are associated with severe health risks like organ damage, electrolyte imbalances, and long-term gastrointestinal complications [8]. Stimulants also have fast-acting effects, which may reinforce the misguided belief that they can prevent calorie absorption by inducing diarrhoea [9].

Understanding laxative misuse among individuals with eating disorders, and more broadly, laxative use within the general population [1], has primarily relied on self-reported data [3,4] which is subject to recall errors and social desirability effects [10]. Given the stigma attached to such usage, supermarket loyalty card data promise a non-invasive and relatively unexplored method for tracking self-medication, having only been applied in this setting in a handful of studies. Loyalty cards are being increasingly used within health research [11], capturing objective individual

transactions over many months or years. By aggregating data at a population level, these novel datasets offer granular insights at high coverage.

Previous studies have explored the predictive power of over-the-counter medication sales in assessing common ailments in the UK: pain relief and cough and cold medicines have been directly linked to respiratory mortality rates at local levels [12]; the impact of demographics in predicting treatment of minor ailments has been examined, alongside sun preparation sales, with men making far fewer purchases than women [13]; and purchases of "symptomatic" over-the-counter medications (e.g., indigestion medicines) have been shown to be indicative of subsequent ovarian cancer diagnoses [14]. No studies have yet used loyalty card transactions to track laxative self-medication, reflecting a potential gap given the significant risks associated with misuse of such products. We address that gap by exploring patterns in laxative self-medication, with a focus on variations during the New Year period and over the summer months, in order to investigate instances where changes in use might be related to weight loss attempts.

### 1.1. Hypotheses

Loyalty card data are examined to identify seasonal patterns in laxative purchasing, hypothesising that stimulant laxative purchases in relation to intended weight loss will display two primary seasonal peaks: one following the festive season, reflecting "New Year's resolutions" around weight loss, and another during the summer, a period often associated with heightened attention to appearance due to the warmer weather.

For the New Year period, we propose that:

• **H1a)** the number of stimulant laxative doses purchased will be higher in January than in the December prior;

• **H2a)** this seasonal variation will be more evident for stimulant laxatives than for other types of laxatives;

• **H3a)** this seasonal variation will be also observed in "diet and weight management" products sold at the same retailer.

We anticipate a second seasonal increase in the warmer months, and hypothesise that:

• **H1b)** the number of stimulant laxative doses purchased will be higher in May, June, July, and August than in September;

• **H2b)** this seasonal variation will be more evident for stimulant laxatives than for other types of laxatives;

• **H3b)** this seasonal variation can also be observed in "diet and weight management" products sold at the same retailer.

We expect that laxative use to treat constipation will show little seasonal variation, as constipation is well evidenced to not vary seasonally [15]. However, we do expect laxative use with the goal of weight loss to vary seasonally in line with other indicators of attempted weight loss (e.g., purchasing fewer calories) [16]. As such, we assume that seasonal variation in laxative purchases is likely related to their use for attempted weight loss.

Negative controls of other products are used to observe whether seasonal variation is due to other unrelated factors. That is, we expect that other groups of medication (pain relief, hay fever, cold and flu) will not show the same seasonal pattern in sales, as need for these products vary throughout the year. With pain relief products, we do not expect any seasonality; with hay fever, we expect increases in spring or early summer due to common rise of pollen-induced allergy symptoms; and with cold and flu we expect an increase in colder months. In addition, shampoo sales are included to capture overall seasonal trends in in-store traffic, as these should otherwise be fairly consistent throughout the year.

## 2. Results

### 2.1. Descriptive summary

Data from the top 10% of laxative buyers were used within our analyses, representing 73,742 customers each purchasing at least 60 doses between December 2013 and December 2014. This accounted for 726,357 items purchased. The

PLOS Digital Health

mean number of doses in a single stimulant product was 21.31 (SD = 15.52) while the mean number of doses in a single non-stimulant product was 8.93 (SD = 6.11). This means that a stimulant product would last for a maximum of 21 days on average, while a non-stimulant product would last maximum 9 days.

**2.1.1. Monthly product sales.** Mean product sales per month were calculated between December 2013 and December 2014 for the top 10% of buyers in six product categories: laxatives (n = 85,578 buyers), pain killers (n = 553,414), weight management (n = 109,827), hay fever (n = 236,085), cold/flu medications (n = 469,651), and shampoo (n = 626,501 buyers). This is shown in Fig 1. Note that actual sales units (i.e., number of products sold) were used here, as it was both infeasible to manually translate all products across all categories into dosages, and shampoo could not be represented in this way.

The trends in Fig 1 are as expected. Specifically, sales of hay fever products peak in spring in line with seasonal allergies, cold and flu medication sales peak in the winter months in line with seasonal illnesses, pain product sales remain relatively consistent (with some increase during winter), and weight management product sales increase in the summer months and in the New Year (with a dip in February). Shampoo sales fluctuate throughout the year, with peaks in March, May/June, and September. For laxatives, the trends are less clear, with fluctuations year-round. We also plot monthly doses purchased for different laxative products (see Fig 2), for the top 10% of buyers of all laxatives (n = 85,578 buyers), stimulants (n = 60,685), and non-stimulants (n = 20,109).

For stimulant laxatives, we observe a slow increase in doses purchased through the summer, decreasing again in September, and increasing in the lead up to Christmas; we also see a decrease at the start of the year. For non-stimulants, we see a large spike in doses purchased in July, and an increase around the New Year. Fewer doses of non-stimulants are sold overall.

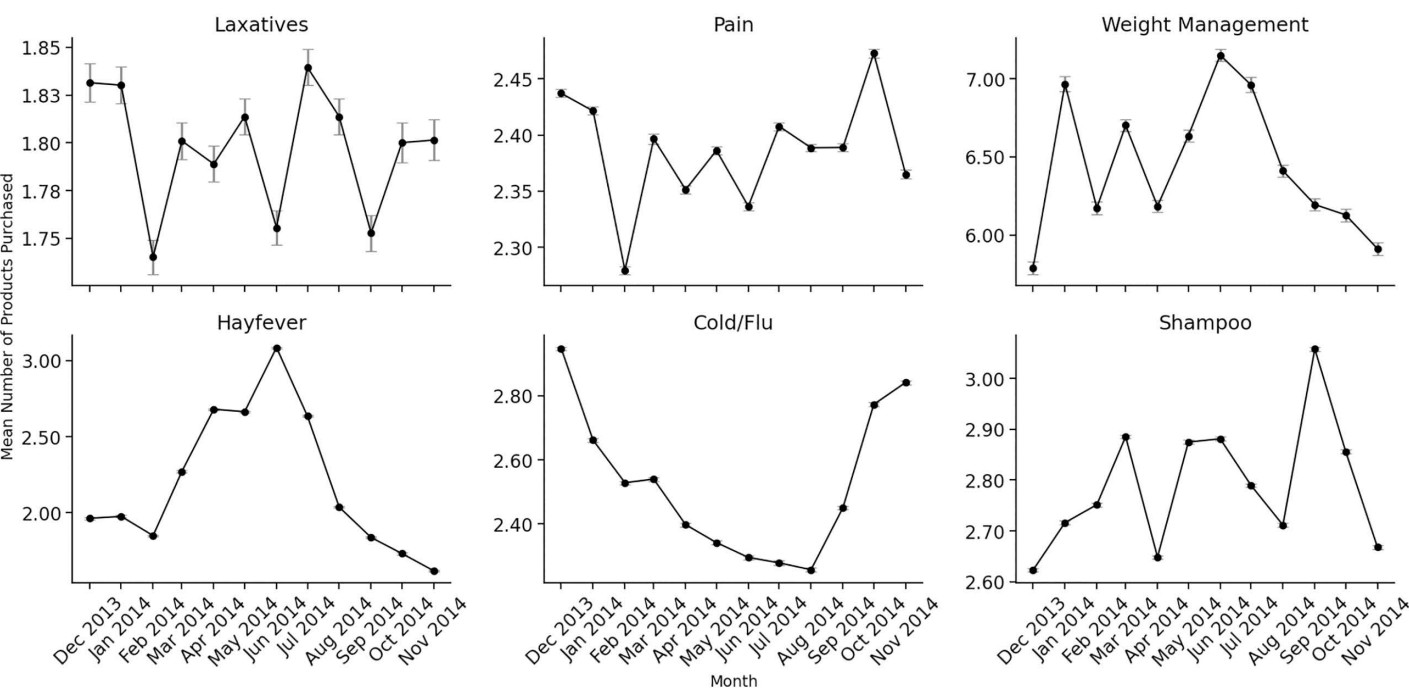

**Fig 1. Average monthly sales units (product sales) by product category from December 2013 to November 2014; intervals demonstrate standard errors.**

PLOS Digital Health

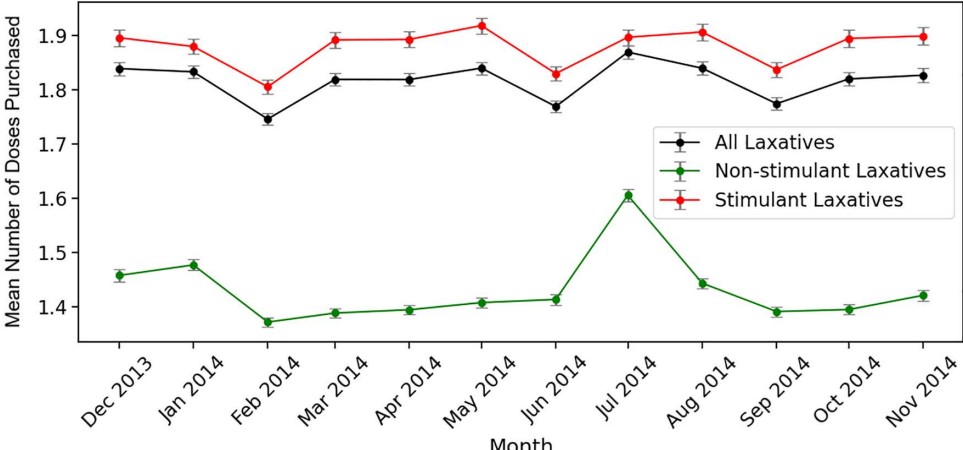

**Fig 2. Average monthly laxative doses by laxative type, purchased from December 2013 to November 2014; intervals demonstrate standard errors.**

## 2.2. Regression analyses

### 2.2.1. Seasonal variation in stimulant laxative purchases.
We first tested whether there was seasonal variation in stimulant purchases, comparing doses purchased in January and December (H1a), as well as in summer months and September (H1b). Full regression results, including coefficients and Incidence Rate Ratios (IRRs), are shown in Table 1.

For H1a, there is no evidence for a difference in stimulant laxative purchasing between December and January. However, stimulant dose purchases were around 2% higher in July, 3% higher in August, and 3% lower in May compared to September (H1b). There was no meaningful difference in June. This partially supports our hypothesis. In both models, the significance of the inflation constant and alpha justify the use of the model.

Model comparisons showed that the ZINB model consistently outperformed standard negative binomial (NB) and Poisson alternatives. For H1a, the ZINB model achieved the lowest AIC/BIC values (AIC = 652,627.40; BIC = 652,664.71)

**Table 1. Regression results for H1a and H1b.**

| Hypothesis | Variable | Coefficient | CI | p | p_adj | IRR | IRR_CI |
|---|---|---|---|---|---|---|---|
| H1a | inflation constant | -0.79 | -0.80, -0.78 | <.001 | – | – | – |
| | intercept | 3.86 | 3.85, 3.88 | <.001 | – | – | – |
| | month (Jan = 1) | 0.00 | -0.01, 0.01 | .856 | – | 1.00 | 0.99, 1.01 |
| | alpha | 34.28 | 32.79, 35.78 | <.001 | – | – | – |
| H1b | inflation constant | 0.05 | 0.04, 0.06 | <.001 | – | – | – |
| | intercept | 3.84 | 3.83, 3.86 | <.001 | – | – | – |
| | May | -0.03*** | -0.03, -0.02 | <.001 | <.001 | 0.97 | 0.97, 0.98 |
| | June | 0.00 | -0.01, 0.00 | .261 | .261 | 1.00 | 0.99, 1.00 |
| | July | 0.02*** | 0.01, 0.03 | <.001 | <.001 | 1.02 | 1.01, 1.03 |
| | August | 0.03*** | 0.02, 0.03 | <.001 | <.001 | 1.03 | 1.02, 1.03 |
| | alpha | 33.39 | 31.98, 34.81 | <.001 | – | – | – |

*<.05, **<.01, ***<.001. p_adj refers to Holm-adjusted p values; IRR refers to Incidence Rate Ratios; IRR_CI refers to the 95% confidence interval for the IRR.

relative to NB (AIC = 684,920.99; BIC = 684,948.98) and Poisson (AIC = 4,841,093.91; BIC = 4,841,112.57). For H1b, the same pattern was observed (ZINB AIC = 1,783,666.11; BIC = 1,783,740.27; NB AIC = 1,895,763.84; BIC = 1,895,827.40; Poisson AIC = 16,594,493.46; BIC = 16,594,546.43). Vuong tests further supported the ZINB model (H1a: Z = 73.64 vs NB, 61.42 vs Poisson; H1b: Z = 134.37 vs NB, 118.43 vs Poisson; all p < .001), confirming that inclusion of a zero-inflation term provided the best fit.

**2.2.2. Seasonal variation in laxative purchases, modulated by laxative type.** We tested whether seasonal variation in laxative purchasing was dependent on the laxative type (stimulant versus non-stimulant), comparing doses purchased in January and December (H2a), as well as in summer months and September (H2b). Results are shown in Table 2.

For H2a, stimulant dose purchases were 10% higher in January than in December. In line with Fig 2, doses of stimulant laxatives were purchased far more frequently overall than non-stimulants: approximately 18 times higher on average. However, the seasonal rise was about 9% weaker for stimulants relative to non-stimulants. For H2b, purchases were 20% higher in May and 8% higher in August compared to September. No meaningful differences were observed for June or July. Again, stimulant doses were purchased at far higher levels than non-stimulants. The interaction terms indicate that the seasonal increases in May and August were attenuated for stimulants: the May interaction shows that the seasonal peak was 19% smaller for stimulants, while the August interaction indicates a 5% smaller increase relative to non-stimulants. Together, these results suggest that while stimulant laxatives are purchased at substantially higher baseline levels, seasonal variation in laxative purchasing is more pronounced for non-stimulant products.

Model comparisons showed that the ZINB model consistently outperformed standard NB and Poisson alternatives. For H2a, the ZINB model achieved the lowest AIC/BIC values (AIC = 862,692.59; BIC = 862,753.58) relative to NB (AIC = 902,385.13; BIC = 902,435.96) and Poisson (AIC = 6,591,206.61; BIC = 6,591,247.27). For H2b, the same pattern

**Table 2. Regression results for H2a and H2b.**

| Hypothesis | Variables | Coefficient | CI | p | p_adj | IRR | IRR_CI |
|---|---|---|---|---|---|---|---|
| H2a | Inflation constant | -0.37 | -0.38, -0.35 | <.001 | – | – | – |
| | Intercept | 0.99 | 0.95, 1.03 | <.001 | – | – | – |
| | Month (Jan = 1) | 0.09*** | 0.06, 0.12 | <.001 | – | 1.10 | 1.07, 1.13 |
| | Type (Stim = 1) | 2.89*** | 2.85, 2.92 | <.001 | – | 17.91 | 17.28, 18.56 |
| | Month:Type | -0.09*** | -0.12, -0.06 | <.001 | – | 0.91 | 0.88, 0.94 |
| | Alpha | 36.52 | 35.17, 37.86 | <.001 | – | – | – |
| H2b | Inflation constant | 0.29 | 0.28, 0.30 | <.001 | – | – | – |
| | Intercept | 0.97 | 0.93, 1.01 | <.001 | – | – | – |
| | May | 0.19*** | 0.15, 0.22 | <.001 | <.001 | 1.20 | 1.17, 1.24 |
| | June | 0.01 | -0.02, 0.04 | .607 | .999 | 1.01 | 0.98, 1.04 |
| | July | 0.03 | 0.00, 0.06 | .068 | .270 | 1.03 | 1.00, 1.06 |
| | August | 0.08*** | 0.05, 0.11 | <.001 | <.001 | 1.08 | 1.05, 1.12 |
| | Type (Stim = 1) | 2.89*** | 2.85, 2.92 | <.001 | <.001 | 17.96 | 17.33, 18.62 |
| | May:Type | -0.21*** | -0.24, -0.18 | <.001 | <.001 | 0.81 | 0.78, 0.84 |
| | June:Type | -0.01 | -0.05, 0.02 | .443 | .999 | 0.99 | 0.96, 1.02 |
| | July:Type | -0.01 | -0.04, 0.02 | .576 | .999 | 0.99 | 0.96, 1.02 |
| | August:Type | -0.05** | -0.08, -0.02 | .001 | .006 | 0.95 | 0.92, 0.98 |
| | Alpha | 35.46 | 34.19, 36.73 | <.001 | – | – | – |

*<.05, **<.01, ***<.001. p_adj refers to Holm-adjusted p values; IRR refers to Incidence Rate Ratios; IRR_CI refers to the 95% confidence interval for the IRR.

was observed (ZINB AIC = 2,311,583.12; BIC = 2,311,719.97; NB AIC = 2,435,379.42; BIC = 2,435,504.87; Poisson AIC = 20,838,585.67; BIC = 20,838,699.71). Vuong tests further confirmed superior fit for ZINB (H2a: Z = 92.03 vs NB, 78.04 vs Poisson; H2b: Z = 157.17 vs NB, 139.72 vs Poisson; all p < .001). Together, these results confirm that including a zero-inflation term provided the most accurate representation of the data across both models.

**2.2.3. Seasonal variation in weight management product purchases.** Finally, we tested whether there was seasonal variation in general diet and weight management products, comparing trends between January and December (H3a), as well as between the summer months and September (H3b). This was tested for the top 10% of buyers of weight management products (n = 109,827), covering 3,338,922 purchases.

Of these, 2990 customers also bought a laxative product (of any type). Time-lagged correlations within this group revealed a mixed temporal relationship. Purchases within the same month (lag 0) were moderately positively correlated (r = 0.39), indicating that individuals who bought more weight-management products in a given month also tended to purchase more laxatives during that month. However, correlations at subsequent months showed alternating directions: 1 month (lag 1) was weakly negative (r = -0.16), 2 months (lag 2) was positive (r = 0.31), and 3 months (lag 3) was negative (r = -0.35). These results suggest that co-purchasing behaviour is not consistent over time and may fluctuate across months.

Regression results are shown in Table 3.

For H3a, weight management product purchases were 41% higher in January compared to December. For H3b, purchases were higher in all summer months compared to September: 16% higher in May, 32% higher in June, 30% higher in July, and 9% higher in August. These results support both hypotheses (H3a, H3b).

Model fit comparisons showed that the zero-inflated negative binomial (ZINB) model consistently outperformed standard negative binomial (NB) and Poisson alternatives. For H3a, the ZINB model achieved the lowest AIC/BIC values (AIC = 534,523.34; BIC = 534,561.61) compared with NB (AIC = 534,537.03; BIC = 534,565.74) and Poisson (AIC = 1,024,239.65; BIC = 1,024,258.79). For H3b, the same pattern was observed (ZINB AIC = 1,965,702.85; BIC = 1,965,780.16; NB AIC = 1,973,650.40; BIC = 1,973,716.66; Poisson AIC = 4,399,399.70; BIC = 4,399,454.91). Vuong tests strongly favoured the ZINB specification across both models (H3a: Z = 92.03 vs NB, 78.04 vs Poisson; H3b: Z = 36.12 vs NB, 117.58 vs Poisson; all p < 0.001), confirming that inclusion of a zero-inflation term provided the best fit for these data.

Table 3. Regression results for H3a and H3b.

| Hypothesis | Variable | Coefficient | CI | p | p_adj | IRR | IRR_CI |
|---|---|---|---|---|---|---|---|
| H3a | Inflation constant | -3.89 | -4.50, -3.27 | <.001 | – | – | – |
| | Intercept | 1.33 | 1.32, 1.35 | <.001 | – | – | – |
| | Month (Jan = 1) | 0.34*** | 0.33, 0.36 | <.001 | – | 1.41 | 1.38, 1.43 |
| | Alpha | 7.90 | 7.64, 8.15 | <.001 | – | – | – |
| H3b | Inflation constant | -0.79 | -0.82, -0.77 | <.001 | – | – | – |
| | Intercept | 1.44 | 1.43, 1.45 | <.001 | – | – | – |
| | May | 0.15*** | 0.13, 0.16 | <.001 | <.001 | 1.16 | 1.14, 1.18 |
| | June | 0.28*** | 0.27, 0.29 | <.001 | <.001 | 1.32 | 1.30, 1.34 |
| | July | 0.26*** | 0.25, 0.27 | <.001 | <.001 | 1.30 | 1.28, 1.31 |
| | August | 0.09*** | 0.08, 0.10 | <.001 | <.001 | 1.09 | 1.08, 1.11 |
| | Alpha | 7.97 | 7.78, 8.16 | <.001 | – | – | – |

*< .05, **< .01, ***< .001. p_adj refers to Holm-adjusted p values; IRR refers to Incidence Rate Ratios; IRR_CI refers to the 95% confidence interval for the IRR.

## 3. Discussion

This study examined seasonal patterns in laxative purchasing, focusing on whether trends align with seasonal motivations for weight loss and sales of weight management products. Using large-scale, real-world transaction data from a major UK pharmacy retailer over a one year period, we explored how laxative use varied across the New Year and summer months. Laxatives were categorised into stimulant or non-stimulant products based on their active ingredients, and *maximum dose duration* was calculated for individual products to estimate the number of doses purchased at an individual-level. Analyses focused on the top 10% of buyers to identify habitual use.

We hypothesised that stimulant laxative purchases would increase in January compared to December (H1a), reflecting New Year body image concerns, and that this trend would be stronger for stimulants compared to non-stimulants (H2a) due to their perceived weight loss effects. Although we found no difference between doses purchased in January and the December prior with a simple comparison, sales were higher in January when controlling for laxative type. This increase was more pronounced for non-stimulant laxatives, suggesting that stimulant use was less sensitive to monthly change. We also hypothesised increased stimulant purchases during summer versus September (H1b), with a stronger effect for stimulants (H2b). More stimulant doses were purchased in July and August compared to September, although fewer were purchased in May. After controlling for laxative type, the August effect persisted, but the July effect disappeared and the May effect reversed. Again, this trend was stronger for non-stimulants. Together, these results reflect seasonal increases in laxative purchases during January (compared to December) and August (compared to September), which may suggest motivations linked to body image. During the summer, social pressures and seasonal activities associated with warmer weather heighten body dissatisfaction and weight-loss motivation [17], while the New Year brings similar societal expectations to lose weight [18]. However, while our evidence is suggestive of a link, we cannot confirm intent.

We also predicted similar seasonal trends in weight management products. Purchases increased in January compared to December (H3a), and were higher in summer compared to September (with strongest effects in June and July). This aligns with with seasonal weight loss attempts reported in the literature [16], and may reflect the increased focus on body image as temperatures rise [17]. As these patterns mirror those seen in laxatives, this may suggest that both categories are influenced by motivations of weight control.

However, there are notable alternative explanations that cannot be ruled out, as purchasing of laxative and weight management products may be influenced by unrelated health (travel-related constipation, gastrointestinal infections) or retail (in-store traffic, promotions) factors. Even so, our negative controls provide strong evidence that the observed fluctuations are not simply reflections of broader seasonal or retail-driven patterns. Hay fever and cold/flu medications followed expected peaks in winter and spring, respectively, confirming that our approach is sensitive to known seasonal effects. Painkillers showed a modest winter increase, contrary to our expectation of minimal seasonality; but this is consistent with known rises in winter illnesses and therefore does not weaken our interpretation. Shampoo, included as a non–health-related control, shows no seasonality and instead fluctuates randomly throughout the year, typical of general consumer behaviour.

Together, these controls suggest that products known to show predictable seasonality do so in our data, and those without expected seasonality do not. This strengthens the inference that the distinctive patterns in laxative and weight-management purchasing are not attributable to general retail seasonality or broad health-related cycles. Nevertheless, these controls cannot eliminate all alternative explanations, particularly those specific to gastrointestinal conditions or behaviours.

### 3.1. Dosage vs. sales units

We used *maximum dosage duration* as the outcome variable to account for differences in pack size across laxative products (e.g., 12 vs. 48 tablets), offering a more accurate measure of use than sales quantity. This is particularly important

when investigating usage trends, as it reveals whether consumers are using higher or lower doses of a product regardless of how many units they purchase.

By focusing on dosage rather than sales, we were able to identify seasonal variations that might not be apparent, or as accurate, when looking only at sales. Regression results using sales units (available in the Supplementary Material S3 Text) were largely consistent, supporting the same hypotheses (H1a–H2b). Both outcomes showed increased stimulant laxative purchases in January vs. December (H1a, H2a), with stronger effects for stimulants. Similarly, stimulant use was higher in summer than September (H1b), though specific month effects varied (e.g., a July effect appeared in units sold but not dosage). Overall, we estimate that dosage-based analyses provided a more accurate reflection of purchasing behaviour.

Implementing this approach across all product categories in this study was not feasible due to the large number of individual items (e.g., 447 weight management items, 449 pain relief products). However, future research could benefit from incorporating dosage information to provide more accurate insights into self-medication, potentially leveraging natural language processing techniques to facilitate this on a larger scale.

### 3.2. The use of loyalty card data to inform policy

Recent studies have highlighted the potential of loyalty card transactions for evaluating public health policy. For example, prior studies have assessed the impact of the sugar tax in the UK and Spain [19,20], the effectiveness of a salt reduction campaign in the UK [21], and the impact of alcohol policy reform in Finland [22]. However, no studies have yet used loyalty card data to evaluate policies related to self-medication.

In 2020, the Medicines & Healthcare products Regulatory Agency (MHRA) in the UK introduced new regulations to curb stimulant laxative abuse [23]: reducing pack sizes, adding on-pack warnings about weight loss use, and restricting over-the-counter sales to individuals over 18. While this is considered a regulatory advance in comparison to other countries [24], no data-driven evidence has supported the implementation of these measures; the evidence base at the time largely relied on case reports and clinical insight.

Our study may retrospectively support the rationale for the regulation. We identified seasonal fluctuations in laxative purchases, although there were no strong patterns for stimulants in particular (Fig 2). However, stimulants did show an increase in purchases in summer compared to September (Table 1), a period commonly associated with weight loss efforts. Additionally, 2,990 customers purchased both a weight management and laxative product, with time-lagged correlations indicating potential co-purchasing. However, while these findings are consistent with seasonal patterns that may reflect weight-control behaviours, this should be interpreted cautiously, as we cannot confirm actual use or intent behind purchasing. While seasonal factors such as travel-related illness or dehydration could also influence laxative use, we strengthen our interpretation by comparing with unrelated product categories (Fig 1). Taken together, the data suggest that some laxative use may be plausibly linked to misuse with weight management intentions.

While the evidence is not strong enough to directly validate the MHRA laxative regulation, our findings highlight an opportunity for future studies to evaluate seasonal trends in laxative use post-regulation, thereby assessing the policy impact over time. In addition, this demonstrates how loyalty card data can support public health interventions and provide a scalable resource for evaluating policy measures over time in real-world settings.

### 3.3. Strengths and limitations

Our study addresses an important public health concern, using novel loyalty card data [25] to provide insights into real-world laxative purchasing behaviour across a large sample of consumers. By focusing on the top 10% of buyers, we effectively quantify the behaviours of the most habitual laxative users. Further, by translating our outcome to reflect dosage rather than sales units, we allow for a more accurate reflection of consumption. These methodological choices open up new avenues to use transaction data to understand unobserved health conditions.

Our work includes some limitations. Firstly, our analyses may reflect misalignment between shopping behaviour and consumption. For example, one person may buy a 60-day supply in May, while another buys two 30-day supplies in May and June. Despite identical consumption, this discrepancy could bias interpretations of monthly sales trends; this may be addressed in future work by incorporating time-series models. Moreover, standardising product sales by dosage introduced some assumptions about consumer behaviour. In particular, our approach relied on the *maximum dose duration* for an adult; it is likely that not all consumers adhere to this dosage (e.g., due to age, body weight, or personal preferences), instead using a starting or median dose. However, this assumption provides a consistent metric across time and buyers, allowing us to reliably examine relative seasonal patterns. Future work could consider alternative dosing assumptions where such data are available.

Future studies should incorporate linkages to cross-sectional surveys or medical records to delineate motivations for laxative use. The presence of demographic information will provide ground truth, allowing researchers to more accurately infer conditions which may have influenced laxative purchasing (e.g., eating disorders, digestive issues, travel-related constipation).

The dataset itself also has representational constraints. It covers transactions from a single UK pharmacy retailer between December 2013 and December 2014, and participants in the dataset are largely female (87%). The gender imbalance likely reflects the retailer's customer profile rather than national prevalence of laxative use, but caution is necessary in extrapolating patterns to the broader UK population. In addition, the findings may not generalise to modern purchasing environments (>10 years later) or to other retailers and pharmacies. Likewise, as the data pre-date the 2020 MHRA guidance and the wider expansion of online pharmacies, our results cannot speak directly to current or future impacts. Replication with more recent datasets would help assess whether the observed relationships persist over time and across consumer groups. Comparative studies across countries with different regulatory frameworks and policy interventions could provide additional insights into effective strategies to reduce laxative misuse and identify best practices for public health interventions.

Seasonal retailer activity, including promotions, holiday sales periods, and fluctuations in overall store traffic, may also influence purchasing patterns [25]. Although we could not adjust directly for promotional calendars, our shampoo negative control suggests these factors are unlikely to account for the observed effects in laxative purchasing; the variation in shampoo sales did not mirror those of laxatives, indicating that general store traffic alone is insufficient to explain the peaks. In addition, the impact of promotions and increased purchasing during the festive period is an ongoing concern in loyalty card research. While previous studies have addressed this by excluding these weeks from analyses [26], this approach was infeasible in the present study as December was a key month of interest for our hypotheses; consequently, some results may reflect a small upward bias from holiday-related increases in purchasing. Nonetheless, developing analytic techniques to account for these confounding influences will help provide a more accurate understanding of their impact on the sales of health-focused products.

### 3.4. Conclusion

This study provides novel insights into seasonal trends in laxative purchasing, revealing potential links between laxative use and weight loss intentions, particularly during the summer months. By leveraging large-scale transaction data and focusing on dosage rather than sales units, we offer an objective and precise representation of self-medication patterns. Our findings suggest that stimulant laxative purchases fluctuate in ways similar to weight management products, reinforcing concerns about their misuse. Further research is needed to examine post-regulation purchasing patterns to assess whether the MHRA measures have effectively reduced stimulant laxative misuse. Additionally, integrating transaction data with medical records or demographic surveys could help distinguish between legitimate medical use and weight management motivations, informing targeted interventions to address misuse.

## 4. Materials and methods

Fig 3 shows the research process for this study, including initial and analytic sample size, pre-processing, analysis, and interpretation steps.

### 4.1. Ethics statement

Approval for this study was granted by the NUBS Research Ethics Review Panel at the University of Nottingham (reference number 201829095). The research used loyalty card transaction data provided under a formal data-sharing

**1. Initial Dataset**

2,702,449 customers, 234,938,722 items bought

**2. Pre-processing**

Subset dataset to laxative products only

Remove anomalous data (refunds and high purchases)

Categorise laxative products into stimulants versus non-stimulants

Convert items sales to dosage

Subset to top 10% of customers (by dosage)

**3. Analytic Sample**

73,742 customers, 726,357 items bought

**4. Analysis**

Conduct zero-inflated negative binomial (ZINB) regression to test hypotheses H1a-H3b

Evaluate model fit using Vuong tests (ZINB vs. negative binomial vs. Poisson)

Assess relative model performance using Akaike and Bayesian Information Criteria

**5. Interpretation**

Use 95% confidence intervals and p-values to evaluate meaningful effects

Present incidence rate ratios alongside raw model coefficients

**Fig 3. Steps to demonstrate the research process.**

agreement and non-disclosure agreement between the University of Nottingham and a major UK pharmacy retailer. The dataset contained no directly identifiable personal information; all data were pseudo-anonymised by the retailer prior to transfer, and no identifiers or linkage keys were available to the research team.

The study complied with the UK General Data Protection Regulation (GDPR) and the Data Protection Act 2018. As the data were de-identified with no possibility of re-identifying individuals, informed consent was not required. All analyses were conducted within secure institutional computing environments under the terms of the data governance agreements approved by both the University of Nottingham and the retailer. To completely eliminate any risk of re-identification, outputs are reported only in aggregated form, and no attempt was made to link the loyalty card data to any external sources of personal information.

### 4.2. Transaction data

The dataset used in this study comprises individual customers' purchasing histories from a major UK health and beauty retailer, linked through their loyalty cards at the time of purchase, spanning December 2013 to December 2014. Data covers 2568 stores across the UK. Each transaction includes the number of items purchased, the price, the time and date of purchase, and identifier codes for both the product and store. The records were provided to researchers pseudo-anonymised.

The complete retailer dataset includes 2,702,449 customers (n = 234,938,722 item purchases), approximately 87% of whom are female. Of these, 748,375 participants purchased a laxative product during the specified timeframe, after excluding anomalous purchases. Anomalous purchases were defined as purchases with negative sales units (n = 43,677), which represent refunds, or those with sales units exceeding 20 (n = 4), an unusually high number likely due to error.

### 4.3. Data pre-processing

The dataset was prepared by: [1] categorising laxative products into stimulants versus non-stimulants, [2] calculating the *maximum dosage duration* for each product, and [3] limiting analyses to frequent laxative buyers. The process for this is described below.

#### 4.3.1. Categorising stimulants versus non-stimulants.
Laxative products were categorised into three groups: stimulant (denoted as a 1), non-stimulant (0), or other [2]. There were 105 unique laxative products: 50 stimulant, 47 non-stimulant, and 8 categorised as other/undefined. The retailer provided pre-defined product categories, and we used these classifications to identify laxative products from within the larger dataset.

Products were manually defined according to the primary active ingredient in the product. Stimulant medications were defined as those whose active ingredients were: sodium picosulfate (e.g., Dulcolax liquid), bisacodyl (e.g., Dulcolax tablets, retailer own-brand tablets), and sennoside (e.g., Senna tablets, Senokot products). Non-stimulants were defined as those whose active ingredients were: ispaghula husk (e.g., Fybogel), paraffin (e.g., liquid paraffin), lactulose, sterculia (e.g., Normacol), sodium citrate (e.g., Micralax), docusate sodium (e.g., DulcoEase capsules), macrogol (e.g., Movicol sachets), guarana (e.g., supplement), and figs (e.g., syrup of figs supplement). It should be noted that a small number of products had ambiguous classifications (e.g., syrup of figs, guarana), and while included as laxatives under the retailer's classification system, may also be considered supplements. Yet, as these products did not contain a stimulant ingredient, they were classified within the non-stimulant group according to our pre-defined criteria.

The "other" category included non-medication products (like suppository applicators), as well as those whose description was not sufficient to identify the product and its primary ingredient; these products were excluded from analyses.

#### 4.3.2. Defining maximum dosage duration.
Previous works have used sales data to track medication purchases over time [13,14]. However, the number of units sold may be misleading due to differences in the number of doses across medication products. Specifically, if an individual purchases only one box of laxatives per month, the transaction data would reflect a single purchase regardless of the quantity contained in the box (e.g., a pack of 12 tablets versus a pack of

24). As a result, an actual increase in laxative intake could go unnoticed, as the transaction data does not account for the variation in dosage or quantity per pack.

To address this, we defined a standardised variable *maximum dosage duration* for all laxative products, which describes the total number of days' worth of medication contained within the product, according to the maximum recommended adult dosage. The measure is defined as:

$$\text{maximum dosage duration (MDD)} = \frac{\text{total product quantity (count or volume)}}{\text{maximum daily dosage (for an adult)}}$$

Total product quantity was provided within the item descriptions in the transaction data, while dosage information was collected via manual searches by one researcher (R.B.). We used a variety of databases to inform our research, including: the retailer's own website, product specific websites (e.g., https://www.senokot.co.uk/), and online pharmacies (e.g., chemist-4-u). Table 4 provides some example products with their quantity, maximum daily dosage, and *maximum dosage duration*. An extended product code book is provided in the Supplementary Material S1 Text, including the active ingredient in the product, the stimulant classification, and dosage information (with source).

Standardising dosage in this way also allows comparison between different forms of the same medication (e.g., between liquid and tablets), and could be extended to enable comparisons between entirely different types of medications (e.g., painkillers against flu medications).

It should be noted that while dosage (rather than sales units) is used in this study to represent laxative purchasing, this was only possible given the small number of products in the dataset. This allowed for manual labelling of individual products. Ideally, this would have been replicated in our control categories, but was not possible due to the size of those datasets; sale units were instead used in these cases. However, this does not compromise the validity of the comparison, as sales patterns across different medication types are used only to contextualise the laxative data at an aggregate level.

**4.3.3. Frequent buyers as a proxy for habitual users.** Previous studies have highlighted the need to identify regular customers when evaluating transaction data, given that they likely demonstrate more consistent purchasing behaviour [27]. This consistency helps reduce noise in analyses by avoiding one-off or irregular purchases, which may be less informative for understanding temporal trends.

Previous approaches include limiting analyses to participants with self-reported loyalty above 41% [28,29], incorporating loyalty as a control variable [30], or refining datasets to frequent shoppers based on visits or weekly expenditure [31]. In the context of laxatives, frequent buyers may reflect habitual purchasing for ongoing health concerns (e.g., digestive issues), or, in some cases, misuse related to weight management, particularly for stimulant laxatives.

For our primary analyses, we defined frequent buyers as the top 10% of purchasers based on doses purchased between December 2013 and December 2014. This threshold was chosen a priori to balance sample size with capturing individuals with the highest purchasing behaviour, who are most likely to include potential misuse cases. To evaluate the robustness of this choice, we conducted sensitivity analyses using alternative thresholds (top 5% and 20% of buyers), as well as analyses including all buyers (see S2 Text). We also provide descriptive statistics in each case comparing included and excluded buyers. These sensitivity checks confirmed that the direction and general pattern of seasonal effects were consistent

**Table 4. Example product categorisation and dose quantities.**

| Item description | Active ingredient | Stimulant (0=No, 1=Yes) | Quantity (tablet) | Quantity (ml) | Maximum Daily Dosage (tab or ml) | Maximum Dosage Duration |
|---|---|---|---|---|---|---|
| Senokot Dual Relief Tablets 20's G | Sennoside | 1 | 20 | – | 2 (tabs) | 10 |
| FYBOGEL HIFI OR 30 (G) | Ispaghula husk | 0 | 30 | | 2 (tabs) | 15 |
| … | … | … | … | … | … | … |
| SENOKOT SYRUP 150ML (G) | Sennoside | 1 | – | 150 | 10 (ml) | 75 |

across all thresholds, however, the top 10% group captured clear seasonal peaks while avoiding dilution of effects seen in the larger sample sizes (i.e., which likely reflected inflated statistical significance due to the large sample size).

## 4.4. Statistical approach

Each hypothesis is tested using a regression analysis, as summarised in Table 5. The outcome for hypotheses H1a–H2b is the *maximum dosage duration* of either stimulant laxatives (H1a, H1b) or all laxative types (H2a, H2b) purchased by an individual within a given month; for simplicity, this is referred to as "doses purchased". For hypotheses H3a–H3b, the outcome is total sales of diet and weight management products.

### 4.4.1. Model specifications.
We use zero-inflated negative binomial (ZINB) regression in this analysis, a model well-suited for count data characterised by overdispersion (variance exceeding the mean) and an excess of zeros, which is present across all outcomes. The model was fit using the ZeroInflatedNegativeBinomialP function in *statsmodels*. It combines:

- a negative binomial component, fitted with a log link function, modelling the expected number of stimulant laxative units purchased; and

- a zero-inflation component, fitted with a logit link function, modelling the probability of a "structural zero" (i.e., cases where no purchase occurs because the customer did not buy a product during that period).

Put simply, the zero-inflation component distinguishes between people who never buy stimulant laxatives, and those who sometimes buy them but had no purchase in a given month. This helps prevent overestimating demand among non-purchasers while still modelling variation among active buyers. An offset term of log(total laxative units purchased) is included to adjust for differences in individual purchasing exposure. The model is estimated using maximum likelihood via the BFGS optimiser with up to 1,000 iterations. To account for repeated observations within individuals, standard errors are clustered at the buyer level.

Model fit is evaluated using Vuong tests comparing the ZINB model against standard negative binomial and Poisson models. Relative model performance is also assessed using Akaike (AIC) and Bayesian (BIC) information criteria, with lower values indicating better fit.

### 4.4.2. Reporting.
Model outputs were interpreted using 95% confidence intervals and p-values to evaluate any meaningful effects. For ease of interpretation, all regression tables report incidence rate ratios (IRRs) alongside the raw

**Table 5. Hypotheses with regression models and explanations.**

| Hypothesis | Regression | Description |
|---|---|---|
| H1a | Doses of Stimulants Purchased ~ Month | The dependent variable is the number of stimulant laxative doses purchased. The month is represented by a binary variable: December = 0, January = 1. |
| H1b | Doses of Stimulants Purchased ~ May + June + July + August | The dependent variable is the number of stimulant laxative doses purchased. Dummy variables represent each summer month, with September as the baseline. |
| H2a | Doses of Laxatives Purchased ~ Month * Laxative Type | The dependent variable is the number of laxative doses purchased (stimulant or non-stimulant). Month is a binary variable (December = 0, January = 1). A binary categorical variable differentiates stimulant vs. non-stimulant laxatives. An interaction term investigates whether seasonal differences depend on laxative type. |
| H2b | Doses of Laxatives Purchased ~ (May + June + July + August) * Laxative Type | The dependent variable is the number of laxative doses purchased (stimulant or non-stimulant). Dummy variables represent each summer month, with September as the baseline. An interaction term investigates seasonal differences based on laxative type. |
| H3a | Sales of Weight Management Products ~ Month | The dependent variable is the number of "diet and weight management" products sold. Month is represented as a binary variable: December = 0, January = 1. |
| H3b | Sales of Weight Management Products ~ May + June + July + August | The dependent variable is the number of "diet and weight management" products sold. Dummy variables represent each summer month, with September as the baseline. |

model coefficients. Because the count component of the ZINB model uses a log link function, each coefficient represents a proportional change in the expected number of doses purchased. We therefore exponentiated the coefficients to obtain IRRs, which express the change in expected purchases associated with each predictor:

- IRR > 1 indicates an increase in expected dose purchases (e.g., IRR = 1.03 represents a 3% increase in doses purchased);

- IRR < 1 indicates a decrease in expected dose purchases (e.g., IRR = 0.97 represents a 3% decrease in doses purchased).

For hypotheses involving multiple month contrasts (i.e., H1b, H2b, H3b), we applied Holm corrections to control the family-wise error rate, and report p-values for these adjusted values. All other contrasts were pre-specified single comparisons and therefore did not require multiplicity adjustment.

**4.4.3.. Additional analyses.** To explore temporal associations between weight-management and laxative purchases, we performed time-lagged correlation analyses. We first aggregated purchases by month for the top 10% of weight-management product buyers. Monthly purchase totals for weight-management and laxative products were then aligned, and Pearson correlation coefficients were calculated at lags of 0–3 months. Lag 0 represents correlations within the same month, while lag 1–3 represent correlations with weight-management purchases leading laxative purchases by 1–3 months. This approach allowed us to assess whether higher purchases of weight-management products predicted increased laxative purchases in the same or subsequent months.

We also repeated analyses for hypotheses H1a-H2b using sales units as the outcome, to compare against results using dosage (see S3 Text).

**4.4.4. Pre-registration.** Our hypotheses have been pre-registered [32]. We have made some changes to the methodologies which are noted here:

- We use data from 2013-2014, rather than 2014–2015. This is because, upon inspection, data from 2015 was incomplete. To ensure the use of the most recent and complete dataset, we revised the study period.

- We convert product data into dosage (as opposed to sales) to more accurately capture use, as dosage provides a standardised measure that better reflects potential consumption across products of varying sizes.

- We refine analyses to frequent shoppers only (top 10%). This decision improves the reliability of behavioural inferences, as frequent shoppers offer denser, more consistent data. In contrast, infrequent shoppers may introduce noise due to irregular purchasing or external factors (e.g., shopping elsewhere).

We do not believe these changes to have significantly impacted our conclusions.

## Supporting information

**S1 Text. Laxative Product Code Book.**
(DOCX)

**S2 Text. Comparisons of Different Buyer Groups.**
(DOCX)

**S3 Text. Results using Sales Units.**
(DOCX)

## Author contributions

**Conceptualization:** Neo Poon, Edward Hunter Sloan, James Goulding, Helen Bould, Anya Skatova.

**Data curation:** James Goulding.

**Formal analysis:** Romana Burgess.

**Funding acquisition:** Anya Skatova.

**Investigation:** Romana Burgess, Edward Hunter Sloan.

**Methodology:** Neo Poon.

**Project administration:** Anya Skatova.

**Resources:** James Goulding.

**Software:** Romana Burgess.

**Supervision:** James Goulding, Helen Bould, Anya Skatova.

**Validation:** Neo Poon.

**Visualization:** Romana Burgess.

**Writing – original draft:** Romana Burgess.

**Writing – review & editing:** Romana Burgess, Neo Poon, Edward Hunter Sloan, James Goulding, Helen Bould, Anya Skatova.

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
