## [Decision Letter · Decision Letter 0]

19 Oct 2025

Response to Reviewers
Revised Manuscript with Track Changes
Manuscript
**Journal Requirements:**

1. Please ensure that your Ethics Statement is available in its entirety at the beginning of your Methods section, under a subheading 'Ethics Statement'.

2. Please upload separate figure files in .tif or .eps format. Also, remove the figures from your manuscript file but keep the legends.

3. We notice that your supplementary tables are included in the manuscript file. Please remove them and upload them with the file type 'Supporting Information'. Please ensure that each Supporting Information file has a legend listed in the manuscript after the references list.

4. For studies involving third-party data, we encourage authors to share any data specific to their analyses that they can legally distribute. PLOS recognizes, however, that authors may be using third-party data they do not have the rights to share. When third-party data cannot be publicly shared, authors must provide all information necessary for interested researchers to apply to gain access to the data. (https://journals.plos.org/plosone/s/data-availability#loc-acceptable-data-access-restrictions)

**Additional Editor Comments (if provided):**
**Reviewers' Comments:**

The manuscript analyzes 2013–2014 UK loyalty-card transactions from a national pharmacy chain to test existing hypotheses about seasonal variations in laxative purchases, specifically the claimed peaks in January (“New Year’s resolutions”) and summer (“appearance”). It also compares patterns between stimulant and non-stimulant laxatives and with weight-management products. The authors use zero-inflated negative binomial regression (ZINB) with a dosage-based outcome for laxatives (maximum dosage duration) and sales-unit outcomes for comparison categories, focusing on the top 10% of laxative buyers, with additional analyses on all buyers.

Major comments

1. Causal language and strength of inference: The paper often shifts from seasonality and weight-control motivation. While the triangulation with weight-management product seasonality is suggestive, alternative explanations (such as GI infection seasonality, travel-related constipation, hydration/heat effects, promotions, and store traffic) remain plausible. Please temper causal language (e.g., “aligned with,” “may suggest”) throughout and expand the discussion of alternative mechanisms and how your negative controls do (and do not) rule them out. Consider falsification tests using additional OTC categories with similar price/promo dynamics but no body-image link.

2. Sampling frame: “top 10%” buyers: Limiting primary analyses to the top decile of buyers risks selection on the dependent variable and may overrepresent chronic constipation or idiosyncratic shoppers. Provide a priori justification for the 10% threshold, and a data-driven sensitivity grid (e.g., top 5/10/20/30%) showing whether seasonal IRRs are stable across cutoffs. Buyer-level descriptives (e.g., median monthly purchases, interquartile ranges) for the included vs excluded populations. Clear motivation for why “frequent buyers” better represent the mechanisms of interest, given the hypothesis focuses on seasonal (time-varying) behavior, not high levels of purchasing.

3. Modeling strategy & panel structure: You have longitudinal, person-month counts with many zeros. Should the generalized mixed effect model or GEE models need to be used to account for within-subject correlation?

4. Cluster standard errors at the buyer (and possibly store-region) level. If retaining ZINB, specify (i) the variables in the zero-inflation equation, (ii) goodness-of-fit vs NB/Poisson (Vuong, AIC/BIC), and (iii) whether zero inflation is buyer- or structure-driven. Report incidence rate ratios (IRRs) with CIs in tables/figures; they are much easier to interpret than raw coefficients.

5. Controls for promotions and holidays: OTC sales are highly responsive to promotions, leaflets, and seasonal events (e.g., Christmas, summer travel). Please indicate whether you can proxy for these and discuss promotion confounding in the limitations section.

6. Product classification and dosage assumptions: The stimulant/non-stimulant taxonomy is hand-labeled. Provide a codebook (supplement) listing each SKU, active ingredient(s), assigned class, quantity, and dosage assumptions, plus inter-rater checks if applicable. Some items listed as non-stimulant (e.g., guarana, syrup of figs) may be supplements or mixed-purpose—please justify inclusion/exclusion and test robustness, excluding ambiguous products. The maximum adult dose assumption could overstate “doses” vs typical use. Add sensitivity analyses using the recommended starting dose and median dose assumptions where available.

7. Negative controls and expected seasonality: The text states little/no seasonality expected for painkillers, but Fig. 1 (as described) suggests some winter increase; reconcile the expectation with observed patterns and discuss whether that weakens/strengthens your argument. Will adding an unrelated personal-care category (e.g., shampoo) as a “pure retail” control for store traffic seasonality be helpful?

8. Weight-management co-purchase analysis: The reported Pearson correlation (r≈0.11 among overlapping buyers) is weak and ignores time order.

9. Generalizability & time period: Data are from Dec 2013–Dec 2014 and from a single retailer with ~87% female cardholders. Discuss how this affects external validity (gender skew; chain clientele), and whether patterns persist in later years. Please temper any policy-impact language accordingly.

Reviewer #2: Thank you for the opportunity to review this manuscript. Overall, this is a well-structured study that addresses an important public health concern, particularly in the context of eating disorders and self-medication. The manuscript is generally well-written, methodologically rigorous, and offers valuable insights. It represents a strong and original contribution to digital health research, with the potential to inform both clinical practice and public health policy.

The study has several notable strengths. The use of large-scale, real-world transaction data is novel and provides an objective measure of purchasing behaviour. The hypotheses are clearly pre-registered, and the authors have thoughtfully compared laxative purchases with both positive (weight management products) and negative controls (painkillers, cold/flu, hay fever). The decision to examine dosage rather than sales units is a methodological strength, offering a more accurate reflection of potential consumption. Finally, the discussion of policy implications, particularly in relation to MHRA regulations, is relevant and timely.

At the same time, there are areas where clarification and refinement would strengthen the manuscript:

1. The results indicate stronger seasonal variation in non-stimulant laxatives, whereas the hypotheses anticipated stronger effects for stimulant products. While this discrepancy is acknowledged, the introduction could be adjusted to recognize that consumer behaviors may not follow mechanistic assumptions about stimulant misuse. This would better frame the findings.

2. Restricting the main analysis to the top 10% of buyers is a defensible strategy, but the rationale for selecting this specific threshold should be more clearly explained. Sensitivity analyses using alternative cut-offs (e.g., top 5%, top 25%) would strengthen the robustness of the conclusions.

3. At several points, the manuscript implies that seasonal peaks in purchasing reflect "misuse" or "body image concerns". While this is plausible, purchase data alone cannot confirm intent. Expanding discussion of alternative explanations (e.g., travel-related constipation, dietary changes, dehydration) would provide a more balanced interpretation.

4. The use of zero-inflated negative binomial regression is appropriate for these data. However, the interpretation of the inflation component is not clearly explained. Providing a brief, plain-language explanation of how the inflation term contributes to the results would improve accessibility for readers less familiar with this approach.

5. The discussion linking findings to MHRA's stimulant laxative regulations is interesting, but the evidence presented does not show stimulants as the main driver of seasonality. The policy relevance could be reframed more broadly, highlighting how loyalty card data can inform public health interventions and be used to evaluate the impact of regulatory measures over time.

6. The manuscript is generally well-written, though some sentences in the Methods section are long and dense; simplifying these would improve clarity.

7. Figures are informative, but y-axis labels could be made more descriptive to enhance interpretability.

8. The abstract slightly overstates conclusions by attributing observed purchasing patterns directly to body image concerns; this should be softened to reflect the indirect nature of the evidence.

Addressing the points above, particularly clarifying methodological choices, softening causal claims, and refining the presentation; will enhance the clarity and impact of the manuscript. With these adjustments, the paper will make a meaningful addition to the literature on digital health, consumer behavior, and public health policy.

Reviewer #3: Your study addresses an important and novel question by applying large-scale loyalty-card transaction data to investigate seasonal patterns in laxative purchases and their potential link with weight-control behaviours. The use of real-world data and a pre-registered analysis plan are significant strengths, and the statistical approach you employ (zero-inflated negative binomial regression) is broadly appropriate for this type of data. The manuscript is also clearly written, intelligible, and accessible to a wide readership; however, there are several issues that must be addressed before publication. My detailed comments are as follows:

Publication criteria & rigor

While the dataset and analysis have potential, the manuscript in its current form does not yet fully meet PLOS Digital Health’s standards of methodological and ethical rigor. In particular, further detail on data governance, reproducibility, and robustness of findings is required.

Statistical analysis

The choice of modelling framework is appropriate, but important details are missing. Please report model specifications (link function, offsets, clustering by customer or store, diagnostics, handling of multiple comparisons). Present effect sizes in natural units (e.g., % change or mean additional doses purchased) with confidence intervals, in addition to regression coefficients.

Justify the decision to focus on the “top 10%” of frequent buyers and show that findings are robust across alternative thresholds (e.g., top 5%, 20%, and all buyers). At minimum, integrate results for all buyers into the main text.

Data availability and reproducibility

The Data Availability Statement does not currently comply with PLOS policy. While raw data may be restricted by license, you must:

Provide a clear route for other researchers to request access to the dataset under similar terms.

Deposit all analysis code, product categorisation lists, and dosage conversion lookups in a public repository with a DOI.

Provide de-identified or synthetic datasets (or at least the data underlying summary statistics) to ensure analyses can be reproduced.

Ethics and data governance

Please include the ethics approval reference number, details of the retailer data-sharing agreement, and confirmation of GDPR compliance. Clarify what steps were taken to minimise risks of re-identification.

Interpretation of findings

Some statements imply stronger causal conclusions than the data support (e.g., that purchases demonstrate misuse or that the findings directly support MHRA policy). Please temper this language. It would be more accurate to state that the findings are consistent with seasonal patterns that may reflect weight-control behaviours, while acknowledging that intent cannot be confirmed.

Potential confounders

Seasonal retailer promotions, holiday sales, and overall transaction volume could influence purchasing patterns. Please either adjust for these factors statistically or expand the limitations section to quantify their possible impact.

Generalisability and limitations

The dataset is skewed toward female customers and limited to one retailer. Discuss how this affects generalisability. Acknowledge that the data cover 2013–2014 and therefore cannot speak directly to current post-2020 policy impacts.

Clarity and presentation

The manuscript is generally well written in standard English, with only minor grammatical edits required. However, please expand the methods section (particularly product classification and modelling) for clarity and reproducibility. Simplify some long sentences in the discussion for easier readability.

In summary, this study has strong potential and could make a valuable contribution to digital health and public health literature. However, major revisions are required to ensure compliance with PLOS data and ethics policies, strengthen transparency of methods, provide fuller statistical reporting, and temper interpretive claims.

**Figure resubmission:**

**Reproducibility:** To enhance the reproducibility of your results, we recommend that authors of applicable studies deposit laboratory protocols in protocols.io, where a protocol can be assigned its own identifier (DOI) such that it can be cited independently in the future. Additionally, PLOS ONE offers an option to publish peer-reviewed clinical study protocols. Read more information on sharing protocols at https://plos.org/protocols?utm_medium=editorial-email&utm_source=authorletters&utm_campaign=protocols

---

## [Decision Letter · Decision Letter 1]

23 Feb 2026

Response to Reviewers
Revised Manuscript with Track Changes
Manuscript
**Journal Requirements:**
**Additional Editor Comments (if provided):**
**Reviewers' Comments:**

**Comments to the Author**

Reviewer #1: All comments have been addressed

Reviewer #4: (No Response)

Reviewer #5: All comments have been addressed

publication criteria?

Reviewer #1: Yes

Reviewer #4: Partly

Reviewer #5: Yes

3. Has the statistical analysis been performed appropriately and rigorously?

Reviewer #1: Yes

Reviewer #4: No

Reviewer #5: Yes

4. Have the authors made all data underlying the findings in their manuscript fully available (please refer to the Data Availability Statement at the start of the manuscript PDF file)?

Reviewer #1: No

Reviewer #4: No

Reviewer #5: Yes

5. Is the manuscript presented in an intelligible fashion and written in standard English?

Reviewer #1: Yes

Reviewer #4: No

Reviewer #5: Yes

Reviewer #1: Authors addressed all my comments

Reviewer #4: 1.In the abstract, please avoid using parentheses for explanations such as “(i.e., painkillers, cold and flu, hay fever, and shampoo)”.

2.The manuscript contains a large number of em dashes (“—”). Please remove them.

3.Please include all keywords in the abstract.

4.Please replace all occurrences of “We” with “This study.”

5.The reference list is too limited. Please add more recent references from the past few years.

6.Please add a study workflow flowchart and provide an explanation of the research process.

Reviewer #5: The authors have adequately addressed all comments raised in the previous round of review. Revisions have improved clarity, strengthened methodological transparency, and ensured that conclusions are appropriately framed in relation to the data.

The manuscript is now suitable for publication.

**Do you want your identity to be public for this peer review?** For information about this choice, including consent withdrawal, please see our Privacy Policy

Reviewer #1: No

Reviewer #4: No

Reviewer #5: **Yes:** NOOR AL-DEEN SHEHAB

**Figure resubmission:**

**Reproducibility:** To enhance the reproducibility of your results, we recommend that authors of applicable studies deposit laboratory protocols in protocols.io, where a protocol can be assigned its own identifier (DOI) such that it can be cited independently in the future. Additionally, PLOS ONE offers an option to publish peer-reviewed clinical study protocols. Read more information on sharing protocols at https://plos.org/protocols?utm_medium=editorial-email&utm_source=authorletters&utm_campaign=protocols

---

## [Editor Report · Decision Letter 2]

26 Feb 2026

Using supermarket loyalty card data to investigate seasonal variation in laxative purchases in the UK

PDIG-D-25-00477R2

Dear Dr Burgess,

Congratulations! We are pleased to inform you that your manuscript 'Using supermarket loyalty card data to investigate seasonal variation in laxative purchases in the UK' has been provisionally accepted for publication in PLOS Digital Health.

Best regards,

Tianzhen Chen

Academic Editor

PLOS Digital Health